# ML²B: MultiLingual ML Benchmark For AutoML

## Abstract

Large language models (LLMs) have recently demonstrated strong capabilities in generating machine learning (ML) code, enabling end-to-end pipeline construction from natural language instructions. However, existing benchmarks for ML code generation are mainly restricted to English, overlooking the global and multilingual nature of ML research and practice. To address this gap, we present ML²B, the first benchmark for evaluating multilingual ML code generation. ML²B consists of 35 Kaggle competitions in 13 natural languages, covering tabular, text, and image data types, with structured metadata and validated human-reviewed translations. For evaluation, we employ AIDE, an automated framework for end-to-end assessment of data science pipelines, and provide insights into cross-lingual model performance. Overall, the results indicate that cross-lingual performance remains unstable, even for languages with substantial training data. The benchmark, evaluation framework, and comprehensive results are made available through our GitHub repository to facilitate future research in multilingual ML code generation: https://github.com/AnonimusCoders/ML²B.

## 1 Introduction

Machine learning (ML) has become a fundamental component in a wide range of contemporary tasks across various domains. Motivated by the necessity to relieve ML researchers from the time-consuming task of baseline pipeline selection or to give a working solution for people out of ML, AutoML frameworks have emerged to automate this process (Zöller & Huber, 2021).

At the same time, LLMs have demonstrated remarkable capabilities in generating code for a wide range of ML tasks, spanning data preprocessing, feature engineering, and the construction of complex model architectures (Chen et al., 2021; Roziere et al., 2023; Li et al., 2023). This progress has motivated the development of benchmarks designed to evaluate ML-oriented code generation, such as MLE-Bench (Chan et al., 2025), DA-Code (Huang et al., 2024b), and Weco-Kaggle (Jiang et al., 2025), which leverage real-world Kaggle competitions to assess model performance on end-to-end ML workflows. Complementary to these efforts, several benchmarks focus on evaluating isolated LLM capabilities for specific ML tasks, e.g., DSCodeBench (Ouyang et al., 2025), DS-1000 (Lai et al., 2023), MLAgentBench (Huang et al., 2024a), ML-Dev-Bench (Padigela et al., 2025). Table 1 provides a detailed comparison of recent and widely used ML code-generation benchmarks.

Though these benchmarks are suitable for their prime task, all of them have a limitation of containing data only in English. Jin et al. (2024), and Raihan et al. (2025) have claimed that there is a large gap between LLM performance on English and other languages, especially low-resource ones, and that it is crucial to evaluate LLM performance on different natural languages.

This gap is especially concerning for ML code generation. First, ML research and practice is global, with substantial activity in non-English-speaking regions. Second, ML code generation inherently requires cross-lingual alignment: models must interpret problem descriptions in diverse languages while producing executable code, typically in English. Current benchmarks cannot measure this ability.

Another challenge arises from *benchmark data leakage*, when benchmark data is also present in the LLM training data (Matton et al., 2024). This issue is particularly important, as the model may overperform in particular benchmark tasks. In the worst case scenario, this may lead the affected

| Benchmark | Multilingual | Leakage Prevention | Private Competitions | ML Capabilities | Isolated Grading |
|---|---|---|---|---|---|
| ML²B (ours) | ✓ | ✓ | ✓ | ✗ | ✓ |
| MLE-Bench | ✗ | ✗ | ✗ | ✗ | ✗ |
| ML-Dev-Bench | ✗ | ✗ | ✓ | ✓ | ✗ |
| MLAgentBench | ✗ | ✗ | ✗ | ✓ | ✗ |
| DSCodeBench | ✗ | ✗ | ✗ | ✓ | ✗ |

Table 1: Comparison of ML²B with other related ML benchmarks. Leakage Prevention refers to code leakage detection and prevention measures. Private Tasks refers to tasks for which solutions have not been publicly released, and therefore may not have been included in the training data of LLMs. ML Capabilities refers to the inclusion of tasks that assess specific machine learning capabilities. Isolated Grading refers to verification of the solution in an isolated container without external internet access.

benchmark competitions to be inconclusive (Zhou et al., 2025). A similar form of data leakage happens when unintended information about the test set appears in training data. Leakage can artificially inflate performance and produce unreliable results (Apicella et al., 2025; Yang et al., 2022; Sasse et al., 2025). It is also pervasive in real-world ML code (Kapoor & Narayanan, 2023).

To address these shortcomings, we introduce ML²B (Multilingual Machine Learning Benchmark), the first benchmark for evaluating LLMs on generating complete ML pipelines from multilingual natural language descriptions. ML²B extends real Kaggle competition tasks into 13 languages while preserving the realism and complexity of full ML workflows.

Our key contributions are the following.

1. **Multilingual benchmark:** A curated dataset of 35 Kaggle competitions, translated into 13 natural languages, creating 455 unique evaluation instances.

2. **Private Competitions:** Inclusion of 10 private competitions to mitigate benchmark data leakage. Since the code and discussions for these competitions are not publicly available on Kaggle, they cannot be part of existing LLM training corpora, providing a more reliable test of generalization.

3. **Robust Code Evaluation:** Verification of the generated solution in an isolated environment with data leakage prevention measures: internet access restriction, modular code submission format and static leakage evaluation.

## 2 RELATED WORK

### 2.1 ML CODE GENERATION BENCHMARKS

Recent benchmarks target ML code generation and workflow evaluation. DSCodeBench (Ouyang et al., 2025) and DS-1000 (Lai et al., 2023) collect large numbers of tasks from GitHub and Stack-Overflow, but mainly assess snippet-level code. Full-pipeline benchmarks include DA-Code (Huang et al., 2024b), which uses open datasets, Weco-Kaggle (Jiang et al., 2025) and MLE-bench (Chan et al., 2025), which leverage Kaggle workflows. MLE-bench evaluates LLM agents on 75 Kaggle competitions, assessing agents ability to compete with human participants.

Some benchmarks specifically target agentic capabilities. For example, ML-Dev-Bench (Padigela et al., 2025) comprises 30 predominantly manually constructed tasks spanning six task categories, e.g., debugging, API Integration. MLAgentBench (Huang et al., 2024a) integrates traditional end-to-end ML workflows with research-oriented tasks, providing a total of 13 evaluation scenarios.

### 2.2 MULTILINGUAL CODE DATASETS

Multilingual datasets for code generation remain scarce. MCoNaLa (Wang et al., 2022) consists of intents for code generation, which are further rewritten by human annotators, and code snippets in Python. RoCode (Cosma et al., 2024) offers Romanian programming problems with Python/C++ solutions. MBPP-Translated (Li et al., 2024) extends MBPP to five languages using machine translation. mHumanEval (Raihan et al., 2025) supports 204 languages, with expert translation for 15,

across 25 programming languages. While these datasets highlight multilingual code generation, none target ML pipelines.

## 2.3 IMPACT OF PROMPT LANGUAGE

Several studies show LLM performance depends strongly on prompt language. Bang et al. (2023); Ahuja et al. (2023); Muennighoff et al. (2023), and Raihan et al. (2025) report substantial drops for low-resource languages. Moumoula et al. (2025) analyze 13 programming and 23 natural languages, showing that non-Latin scripts further degrade performance.

## 2.4 AUTOML FRAMEWORKS

A variety of AutoML systems have been developed, employing distinct methodological approaches and yielding results of varying quality. A detailed discussion of these systems is provided in Appendix A.

Although there is a novel approach in AutoML tasks that focuses on code optimization problems rather than traditional hyperparameter and pipeline optimization, it does not face the challenges mentioned above. The AIDE framework (Jiang et al., 2025) exemplifies this approach, functioning as an LLM Agent for machine learning engineering, which uses solution space tree search and iterative refinement. It has been tested on 75 Kaggle competitions and has shown superior results outperforming LightAutoML (Vakhrushev et al., 2022) and OpenHands (Wang et al., 2025)

Nevertheless, this framework might not be so competitive if tested on competitions with no code solutions publicly available. Consequently, we propose to rigorously evaluate the ML²B benchmark within the AIDE framework to clarify its effectiveness under such closed-code conditions.

# 3 THE ML²B BENCHMARK

Unlike Chan et al. (2025), which relies on full descriptions sourced from the "Overview" and "Data" tabs of competition webpages, ML²B provides structured metadata and task descriptions. We argue that the succinct, structured format of competition data may prove more efficient for large language models (LLMs) while retaining essential information for evaluation. Our benchmark contains rich metadata, task descriptions, and multilingual expansions, enabling standardized evaluation of ML code generation.

## 3.1 BENCHMARK TASK SELECTION

**Competition selection** The ML²B benchmark builds upon the Code4ML 2.0 dataset (Trofimova et al., 2024), which provides standardized task descriptions for Kaggle competitions. From this collection, we identify all competitions that satisfy our inclusion criteria and are not older than 2020. Among these eligible competitions, we manually validate and select a subset of 22 tasks for which public code is available on Kaggle. The selection criteria are:

- The competition is closed, so the data and leaderboard are fixed.
- The dataset is downloadable from the Kaggle page.
- The evaluation metric is documented on the competition page.
- The evaluation is reproducible, with all necessary metadata provided.
- The Kaggle submission format is a tabular prediction.

Since Code4ML 2.0 covers competitions up to 2024, we additionally include 3 new public competitions released in 2025, bringing the total number of public competitions to 25 at the time of this paper. To evaluate LLMs on unseen tasks, we further include 10 additional competitions without publicly available code. Their task descriptions are manually summarized based on the information available on Kaggle.

Overall, ML²B includes 35 competitions and will be expanded further. The full list of competitions is provided in Appendix B.

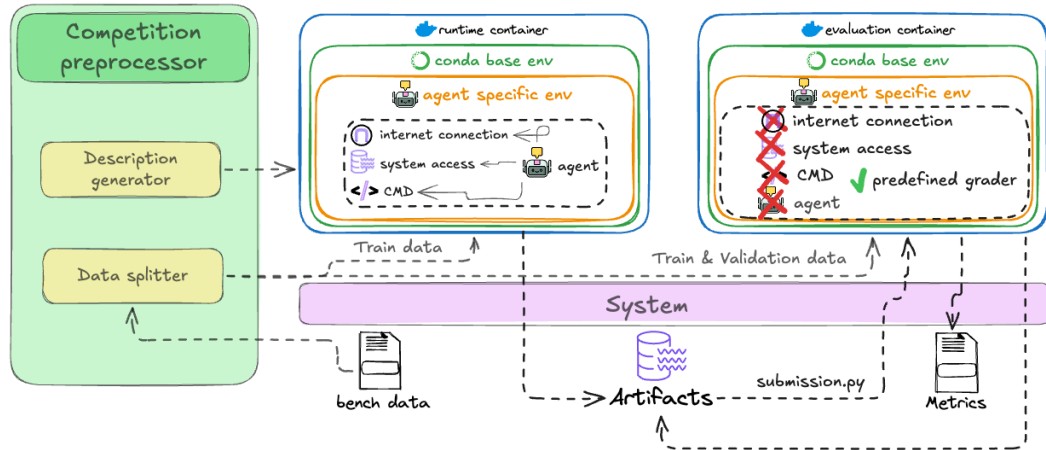

Figure 1: Structure of the ML²B benchmark. Competition preprocessor prepares task descriptions and data, runtime container generates the solution while evaluation container executes the solution code

**Domain coverage and selection**  Each competition in our benchmark has domain information identifying its application area. Domain tags are extracted automatically via an LLM analysis of the data card, description, and competition name. Overall, we cover 12 different domains (see Appendix C).

**Data card standardization**  Unlike Code4ML 2.0, ML²B benchmark includes not only the data source link, but also the data card, which reflects key data information. All data cards are curated manually. Data cards and task descriptions also undergo manual review to ensure clarity and to avoid disclosing information that could give models an unfair advantage. Notably, nearly all Kaggle competition data descriptions include details regarding submission format and test files. However, because test files do not contain target labels, they are irrelevant for our setting, where the framework must produce executable code rather than competition submissions. Therefore, such information is systematically removed. The ML²B benchmark includes information on each task's evaluation metric and its type, mapped according to the scheme proposed by (Drozdova et al., 2023).

## 3.2 METADATA AND STRUCTURE

The benchmark consists of 3 main components (Figure 1), which includes the main competition preprocessor, Docker agent runtime and the submission code grader. Competition preprocessor is responsible for task description generation (see Appendix D) and competition data preparation, the agent runtime manages the AutoML agent, and the code grader evaluates a metric for the submission code in an isolated environment.

### 3.2.1 DOCKER RUNTIME

Both the agent and code grader are executed inside of the Docker environment. The agent Docker image is built from the common runtime image, and both the agent and grader containers are built from the same agent image. This ensures that both the agent and the grader utilize the same Python environment, and at the same time grading is performed in an isolated environment without internet access. This prevents the potentially sensitive evaluation data from leaking in an event of a misconfigured or a malicious script being submitted.

### 3.2.2 CODE GRADER

Instead of the Kaggle-style submission format, which consists of a single submission file, the code grader reproduces the results by executing the submission code directly. Furthermore, the code submitted by the agent must provide specific functions, which are then individually evaluated. Such

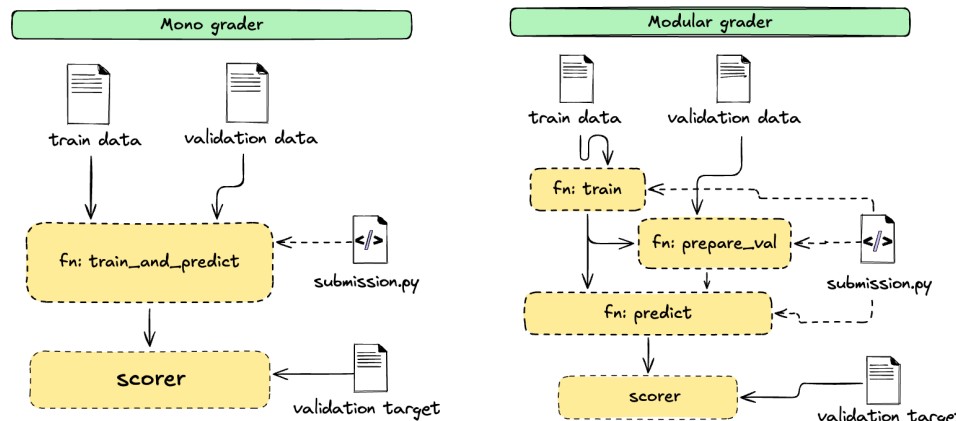

(a) Mono grader is the default code submission format, (b) Modular grader is the introduced format which where a single-function solution returns the prediction isolates the train data from the validation data

Figure 2: Code flow diagram of agent solution submission formats

approach ensures that the submitted code is valid and can be reproduced in the controlled environment. In order to successfully load the submission code, the submission must not have top-level executable code. In order to achieve this, the grader must analyze and recompile the code by performing Abstract Syntax Tree (AST) transformation. Then, the recompiled submission code is executed according to the Section 3.2.3 and the data is loaded into memory by the competition `DataLoader` class. Finally, the resulting submission data is evaluated using the corresponding competition grader function.

### 3.2.3 Submission Code Formats and Data Leakage

The benchmark supports two submission formats: single-function submission format `MONO_PREDICT` (Figure 2a) and modular submission format `MODULAR_PREDICT` (Figure 2b). `MODULAR_PREDICT` consists of three functions `train`, `prepare_val` and `predict`, which sequentially train the model, prepare the prediction data and predict the result. The purpose of such prediction format is to reduce the chance of preprocessing leakage. Preprocessing leakage is the type of data leakage when both training and test data are processed together Yang et al. (2022); Apicella et al. (2025). The most common example is the data normalization being trained on both the training and test features. `MODULAR_PREDICT` format restricts the code flow in a way that the prediction data is introduced only in the second stage of the pipeline, which makes the occurrence of preprocessing leakage less likely. In order to assess the presence of data leakage in submission code, static leakage analysis was performed using the `leakage-analysis` tool (Yang et al., 2022). This tool finds potential relations between the variables and outputs lines of code causing potential data leakage.

### 3.2.4 Data Leakage Assessment

Out of 554 submissions in the `MODULAR_PREDICT` format, 61 (11%) contained potential data leakage according to the tool. By performing further analysis, it was observed that in 8 submissions the data leakage was found in `train` function, which does not operate on prediction data, and in 20 cases the leakage was detected in trivial single-argument functions, which accepted the input data as a single argument (Figure 3). The single-argument function case may be explained as a false-positive, since these functions operated on a single data argument being either the training or prediction data. This leaves the remaining 33 (5.9%) of submissions to have potential data leakage. Overall, the actual data leakage may still be present in the modular submission code if the agent performed model training in the later stages of the code, where both training and prediction data is theoretically accessible. In order to improve the leakage assessment results, further testing using the NBLyzer (Drobnjaković et al., 2024) tool and manual code assessment should be performed.

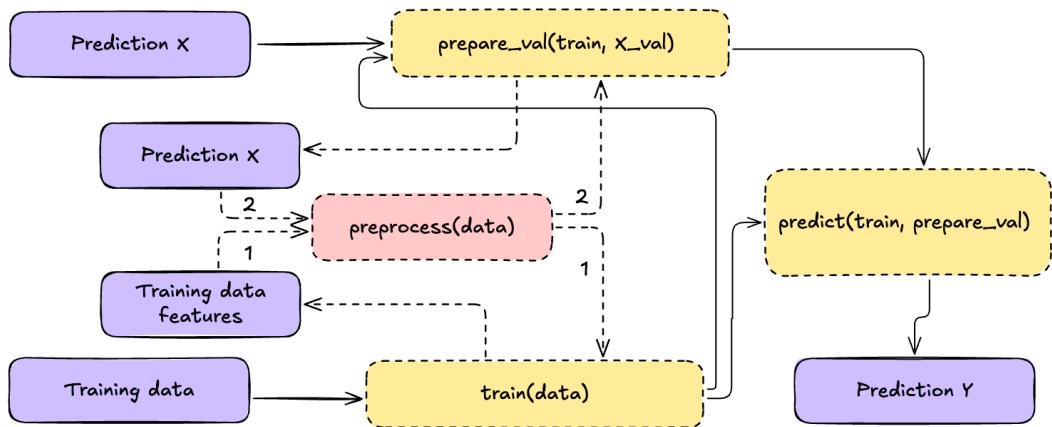

Figure 3: Code flow diagram of a false-positive data leakage, where the preprocessing function is called sequentially

### 3.3 MULTILINGUAL EXPANSION

To obtain a multilingual corpus, we have translated the *domain*, *description*, and *card* fields into target languages (see Appendix E). Other fields do not require translation since they convey universally recognized entities. Following the findings of Jiao et al. (2023), we choose GPT-4o over commercial translators such as Google Translate and DeepL.

After automatic translation, texts in other languages undergo manual review conducted by bilingual annotators with ML experience and at least a bachelor's degree in a computer-science–related field. All annotators participated in the study voluntarily.

To structure validation, each annotator received three Google Forms – one for each translated field. The Form includes the English source, the translation, and a question assessing whether the text sounds natural and preserves the original meaning. If the answer was "No", the annotator supplied a corrected version (see Appendix F). Thus, the final version of the benchmark includes valid translations in Arabic, Belarusian, Chinese, English, Russian, French, Italian, Japanese, Kazakh, Polish, Romanian, Spanish, and Turkish languages. This set of languages is determined by the availability of qualified annotators.

Our goal is not to obtain an idealized or standardized translation, but representative native phrasing. Appendix G reports BLEU-based consistency measurements across languages, while Appendix H summarizes the typical translation patterns observed in GPT-4o outputs.

### 3.4 EVALUATION

To ensure fair comparison of model performance across different competitions, we employ a percentile-based evaluation rather than reporting raw leaderboard metrics. Each model's result is expressed as its percentile rank on the Kaggle public leaderboard, with the 1st percentile indicating top performance and the 100th percentile the weakest. This normalization addresses two issues: (i) competitions use heterogeneous and non-comparable metrics (e.g., RMSE, log-loss, F1-score), and (ii) absolute leaderboard values vary with task design and data scale. Percentiles thus provide a unified, competition-agnostic performance measure that preserves relative standing while mitigating metric-specific biases.

## 4 EXPERIMENTS AND RESULTS

The results reported below are obtained using specific LLM-based agent frameworks selected for the purposes of this benchmark. We do not aim to establish definitive performance rankings of agents or to make broad claims about their universal abilities. Instead, our goal is to provide a cross-lingual

evaluation benchmark that enables systematic analysis of how agentic systems generate end-to-end ML solutions when task descriptions are provided in different natural languages.

Our comprehensive analysis across multiple competitions reveals consistent failure patterns that can be categorized as follows:

- **Missing Training Execution:** Absence of `if __name__ == "__main__"` block prevented model training.
- **Runtime Data Loading:** Attempts to load external data within training functions, violating competition constraints
- **Model Stability:** GPT-4-mini showed higher susceptibility to these errors compared to GPT-OSS variants
- **Inconsistent Preprocessing:** Different feature engineering approaches between training and validation sets
- **Function Signature Modifications:** Despite explicit instructions requiring exact function signatures, agents frequently modified their format
- **Global Dependencies:** Agents consistently violated self-contained code requirements by placing initialization outside function definitions
- **Library and Environment Misalignment:** Systematic use of deprecated API calls and non-existent library functions, the use of non-existent environment library functions

Some of these issues are systematic to particular LLMs, for instance Qwen2.5-coder removed the `Any` keyword import and proceeded to use it later in the code. At the same time, some models like GPT-OSS were less susceptible to these issues. These patterns can be seen in Table 2, where gpt-4.1-mini + deepseek-r1 and gpt-oss-120 + qwen3-coder-30 consistently failed to produce a valid result (see examples in Appendix I)

**Evaluation metric**   We introduce a modified AUP score based on performance profiles (Dolan & Moré, 2004), adapting the approach of Roberts et al. (2023).

For a model $s \in S$ evaluated on tasks $p \in P$, we define the performance profile as:

$$\rho_s(\tau) = \frac{1}{|P|} \left| \{p \in P : r_{p,s} \leq \tau\} \right|,$$

where $t_{p,s}$ is the score of model $s$ on task $p$, $\text{baseline}_p$ is the reference score, and $\tau \geq 0$ is a performance threshold. The relative performance $r_{p,s}$ is defined separately for minimization and maximization tasks:

$$r_{p,s} = \begin{cases} \dfrac{t_{p,s}}{\text{baseline}_p}, & \text{for minimization (lower is better)} \\ \dfrac{\text{baseline}_p}{t_{p,s}}, & \text{for maximization (higher is better)} \end{cases}$$

Thus, $r_{p,s} \leq 1$ indicates that model $s$ outperforms the baseline on task $p$, with smaller values corresponding to better relative performance. The performance profile $\rho_s(\tau)$ represents the fraction of tasks where the model achieves at least a $\tau$-level of performance relative to the baseline.

The performance profile $\rho_s(\tau)$ is a non-decreasing function of $\tau$, starting at $\rho_s(0) = 0$ and approaching $\rho_s(\infty) = 1$. We focus on the region where the model outperforms the baseline ($\tau \in [0, 1]$) and compute the modified AUP score as:

$$\overline{\text{AUP}}_s = \int_0^1 \rho_s(\tau) \, d\tau.$$

This metric aggregates both the proportion of tasks where the model beats the baseline and the magnitude of improvement: a higher $\overline{\text{AUP}}_s$ indicates better overall performance relative to the baseline across all tasks. As the baseline, we use the median score from public Kaggle leaderboard for each task.

Table 2: Sample results of generated ML code validated on the Kaggle platform. Frameworks are executed multiple times per task, as they do not always produce a valid or runnable solution on a single attempt. For each natural language, the AUP score is computed using the best valid submission obtained across these runs. The color scheme reflects the proportion of valid solutions (not necessarily outperforming the baseline): green $\geq 80\%$, yellow $\geq 50\%$, red otherwise. The best results are highlighted in bold.

| Language | AIDE | | | ML-Master | |
| | gpt-oss-120 | gemini-2.5-flash | gpt-4.1-mini | gpt-4.1-mini + deepseek-r1 | gpt-oss-120 + qwen3-coder-30 |
| --- | --- | --- | --- | --- | --- |
| Arabic | 0.071 | 0.080 | 0.076 | **0.102** | 0.034 |
| Belarusian | 0.066 | 0.075 | 0.070 | 0.000 | 0.000 |
| Chinese | 0.073 | 0.074 | 0.072 | 0.063 | 0.000 |
| English | 0.073 | 0.065 | 0.063 | 0.000 | 0.060 |
| French | 0.073 | 0.070 | 0.066 | 0.063 | 0.000 |
| Italian | 0.067 | 0.064 | 0.076 | 0.064 | 0.000 |
| Japanese | **0.118** | 0.065 | 0.071 | 0.021 | **0.061** |
| Kazakh | 0.072 | 0.077 | 0.065 | 0.000 | 0.000 |
| Polish | 0.073 | 0.069 | 0.070 | 0.000 | 0.059 |
| Romanian | 0.066 | **0.080** | 0.070 | 0.000 | 0.061 |
| Russian | 0.067 | 0.068 | 0.067 | 0.000 | 0.061 |
| Spanish | 0.066 | 0.073 | 0.067 | 0.063 | 0.000 |
| Turkish | 0.066 | 0.065 | **0.082** | 0.000 | 0.037 |

**Cross-Lingual Performance Analysis**   Table 2 demonstrates that performance of the models varies systematically across languages. Several cross-lingual tendencies emerge, reflecting differences in language resources and, probably, writing systems. Chinese, English, French, Italian, Japanese, Kazakh, Polish, and Spanish generally achieve high AUP values. Among them Japanese attains the highest individual score (0.118), suggesting highly robust code-generation performance in this language. The persistent presence of these languages in the high-performing cluster provides empirical support for a well-known observation that LLMs exhibit stronger performance on languages that are extensively represented in their pretraining corpora.

However, performance differences do not correlate straightforwardly with writing-system complexity (see Section 2.3). Languages with non-Latin scripts and which are often assumed to be more challenging for tokenization, i.e. Chinese and Japanese, show rather strong performance, likely due to extensive representation in LLM training data and the high degree of lexical borrowing from English in technical and ML terminology, which might reduce ambiguity during code-generation tasks. Conversely, languages such as Arabic, Belarusian, and Russian, despite having simpler tokenization than logographic scripts, do not reach comparable levels of robustness.

Arabic, Belarusian, Romanian, Russian, and Turkish typically occupy the middle range of the distribution. While they frequently yield valid outputs, the results tend to concentrate in the yellow area. These languages often exhibit richer morphology, less standardized ML terminology, or lower representation in pretraining corpora, resulting in decreased stability (Toraman et al., 2023), (Asgari et al., 2025), (Blevins & Zettlemoyer, 2022).

## 5   CONCLUSION

We have introduced ML²B, the first multilingual benchmark for evaluating automated machine learning (AutoML) agents on end-to-end ML pipeline generation from natural language descriptions in multiple languages. Built upon 35 real Kaggle competitions in 13 natural languages and rigorously validated by native speakers, ML²B comprises 455 unique multilingual tasks covering tabular, text, and image data types.

Our systematic evaluation reveals a significant and previously understudied gap in cross-lingual robustness of state-of-the-art LLM-based AutoML agents. Languages with rich morphology and limited representation in pretraining data (e.g., Arabic, Belarusian) exhibit consistently low performance, likely due to lexical sparsity and structural complexity. In contrast, languages with non-Latin

scripts but substantial pretraining presence (e.g., Japanese) show highly unstable performance, suggesting that tokenization issues and script-specific encoding—rather than mere data volume—drive their inconsistency. These findings underscore that English-only benchmarks provide an incomplete and often overly optimistic view of model capability in global ML practice.

To promote rigorous and leakage-aware evaluation, ML²B integrates several methodological advances: (1) inclusion of private competitions to mitigate benchmark data leakage, (2) a modular code submission format designed to reduce preprocessing leakage, and (3) containerized isolated execution to ensure reproducible and secure grading. Our leakage analysis indicates that there exists inadvertent data leakage in generated pipelines.

ML²B serves not only as an evaluation toolkit but also as a diagnostic instrument for identifying language-specific failure modes in ML code generation — such as inconsistent preprocessing, function signature violations, and environmental misalignment — that are invisible in monolingual settings. We open-source the benchmark, evaluation framework, and all associated code to support future research toward more equitable, reliable, and language-inclusive AutoML systems.

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

## A  AUTOML FRAMEWORKS DISCUSSION

There is a large scope of AutoML frameworks that apply different techniques and achieve variable results. For instance, one of the most popular methods involves ML-pipeline and parameter optimization via either Grid Search and Random Search (H2O AutoML (LeDell & Poirier, 2020)) or Bayesian (Auto-sklearn Feurer et al. (2015)) or genetic algorithms (TPOT (Olson et al., 2016)) methods.

One of the most advanced methods in AutoML is Neural Architecture Search (NAS) (Elsken et al., 2019) that automatically designs neural network topologies. Frameworks such as DARTS (Liu et al., 2019) and ENAS (Pham et al., 2018) have shown significant promise in discovering novel, optimized architectures that often outperform manually designed models for specific tasks. It includes three core components: the search space for potential architectures, the optimization methods for discovering the best-performing architecture, and the model evaluation techniques. By automating

the neural architecture design process, NAS can generate more efficient and specialized models, contributing to significant advancements in AutoML.

However, while NAS has achieved remarkable performance, it currently provides limited insights into why certain architectures perform well or how similar architectures are across independent runs. Furthermore, it requires enormous computational resources and accurate design of the search space Liu et al. (2019) that makes it challenging for the ML research.

## B   FULL COMPETITION LIST

The full list of competitions is presented in Table 3

| comp name | Private/Public | Year | data type | Teams | Source |
|---|---|---|---|---|---|
| *WiDS Datathon 2020* | Public | 2020 | tabular | 951 | CODE4ML |
| *IEOR 242 Spring 2020 HW 4* | Public | 2020 | tabular | 72 | CODE4ML |
| *Explicit content detection* | Public | 2020 | text | 106 | CODE4ML |
| *109-1 NTUT Building Deep Learning Applications HW1* | Public | 2020 | image | 90 | CODE4ML |
| *UWaterloo STAT441/841 Data Challenge 1* | Public | 2020 | tabular | 131 | CODE4ML |
| *MADE HW-2* | Public | 2020 | text | 258 | CODE4ML |
| *Financial Engineering Competition (1/3)* | Public | 2020 | tabular | 290 | CODE4ML |
| *Financial Engineering Competition (2/3)* | Public | 2020 | tabular | 268 | CODE4ML |
| *Financial Engineering Competition (3/3)* | Public | 2020 | tabular | 268 | CODE4ML |
| *PRML-Data Contest-Nov 2020* | Public | 2020 – 2021 | tabular | 150 | CODE4ML |
| *Actuarial loss prediction* | Public | 2020 – 2021 | tabular | 140 | CODE4ML |
| *¡She/Hacks¿ - Shaastra'21 and Wells Fargo* | Public | 2021 | tabular | 90 | CODE4ML |
| *ML2021Spring-hw1* | Public | 2021 | tabular | 2032 | CODE4ML |
| *2021-* | Public | 2021 | tabular | 87 | CODE4ML |
| *SYDE 522 (Winter 2021)* | Public | 2021 | tabular | 130 | CODE4ML |
| *Tabular Playground Series - Jul 2021* | Public | 2021 | tabular | 1293 | CODE4ML |
| *Tabular Playground Series - Aug 2021* | Public | 2021 | tabular | 1753 | CODE4ML |
| *Classify Leaves* | Public | 2021 | image | 165 | CODE4ML |
| *Google Brain - Ventilator Pressure Prediction* | Public | 2021 | tabular | 2605 | CODE4ML |
| *Porto Seguro Data Challenge* | Public | 2021 | tabular | 174 | CODE4ML |
| *Crime Learn* | Public | 2021 | tabular | 96 | CODE4ML |
| *Binary Classification with a Tabular Stroke Prediction Dataset* | Public | 2023 | tabular | 770 | CODE4ML |
| *Multi-Class Prediction of Cirrhosis Outcomes* | Private | 2024 | tabular | 108 | CODE4ML |
| *Predicting Optimal Fertilizers* | Public | 2025 | tabular | 2648 | Kaggle |
| *Binary Prediction with a Rainfall Dataset* | Public | 2025 | tabular | 4381 | Kaggle |
| *Eid Al-Adha 2025: Sheep Classification Challenge* | Public | 2025 | image | 355 | Kaggle |
| *Alfa University income prediction* | Private | 2024 | tabular | 8 | Kaggle |
| *2024 DataLab Cup1* | Private | 2024 | text | 108 | Kaggle |
| *Thapar Summer School 2025 — Hack-III* | Private | 2025 | tabular | 110 | Kaggle |
| *Rutgers Data101 Fall2022 Assignment 12* | Private | 2022 | tabular | 162 | Kaggle |
| *CS 506 Fall 2025 Technical Midterm* | Private | 2025 | tabular | 143 | Kaggle |
| *Car Becho Paisa Paao* | Private | 2025 | tabular | 302 | Kaggle |
| *ITMO Flat price prediction 2024* | Private | 2024 – 2025 | tabular | 127 | Kaggle |
| *Multi-label Classification Competition 20* | Private | 2025 | image | 201 | Kaggle |
| *IFT6390-IFT3395: Beer Quality Prediction* | Private | 2025 | tabular | 192 | Kaggle |

Table 3: The full list of competitions included in ML²B

## C   DOMAIN EXTRACTION PROMPT

Figure 4 shows the prompt that we have given to GPT-3.5-turbo model to derive domain tag for each competition. The number of competitions in each domain is presented in Figure 5.

## D   COMPETITION PROMPT EXAMPLES

For constructing the competition prompt, we use a base Markdown template into which the competition-specific task description, data description, and other relevant information are inserted.

Subsequently, depending on the expected structure of the predictions and the type of accepted arguments, a programming-language-specific template is generated for use by the agent.

Listing 1: Base markdown template

```
You will be given a machine learning task description and
    corresponded data path.
```

You are given competition name, data card and description of Kaggle competition. You need to identify the domain that the task belongs to in the given competition.

**Competition name:** Crime_Learn

**Description:** Develop a predictive model to estimate the rate of violent crimes per population in a given area based on specific features. The input consists of two datasets, one for training and one for testing, with the target variable being 'ViolentCrimesPerPop'.

**Data card:** In this competition you will use the sample US crime data for predicting 'ViolentCrimesPerPop'. train.csv – the training dataset.

Figure 4: Example of the prompt used to derive competition domain

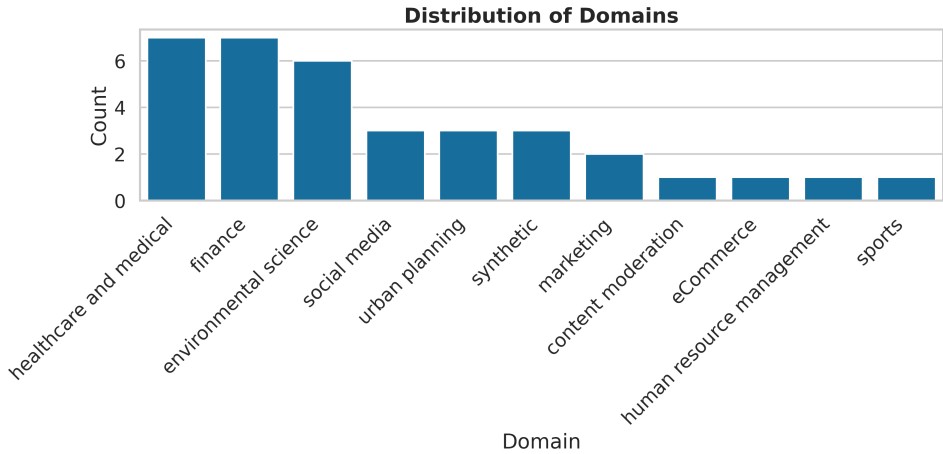

Figure 5: Distribution of competitions over domains

```
Your task is to train a model and generate predictions or
    solutions in one of the programming language: Python, R, Julia
    according to the competition instructions.
You may not manually annotate the data or hardcode outputs for
    specific language.

BENCHMARK INSTRUCTIONS
------
- **First, please read the instructions below which are specific
    to our multilingual benchmark setup.**
- **Submission:** Regardless of the task language, and submission-
    specific instructions you MUST produce a submission at '
    submission/' directory
- **Test set:** As we do not have access to the original test
    labels for some competitions, we have constructed our own train
    /test split. The dataset provided may differ slightly from what
    is described in the competition instructions.
- **Strict programming language requirement:** The programming
    language specified for the task is mandatory. You must
    implement your solution strictly in the given language.
    Solutions written in any other language will not be evaluated.
- **Cheating:** Viewing or copying others multilingual solutions
    is not allowed. Plagiarized submissions will be disqualified.
```

```
Your submission will be graded based on the metric specified in
    the competition specific instructions. Your goal is to achieve
    the best score.
Wherever there is a conflict between the points above and the
    competition-specific instructions above, the above points
    should always take priority since these are always true for our
     offline multilingual benchmark setup.

COMPETITION-SPECIFIC INSTRUCTIONS
------
- **ML task description:** Develop a recommendation system to
    suggest tours to bikers based on various features such as
    bikers' details, friends' status, language preferences, and
    distances between locations. The system should output a list of
     recommended tours for each biker, sorted by the likelihood of
    interest.
- **Task domain** sports
- **Target metric:** map
- **Target solution language** python
- **Data description** train.csv
train.csv has rows corresponding to tours shown to a biker, and
    data about whether he/she liked the tour or not.
biker_id: Unique identifier for a biker.
tour_id: Unique identifier for particular tour.
invited: {0/1} bool variable to denote if the biker was invited to
     the particular tour.
timestamp: Approximate time when the biker was informed about the
    tour.
like: {0/1} bool variable as per the entry made by biker. 1
    indicates biker has liked the tour. 0 indicates that he has not
     responded to the 'like' question.
dislike: {0/1} bool variable as per the entry made by biker. 1
    indicates biker has not-liked the tour. 0 indicates that he has
     not responded to the 'not_like' question.   NOTE : It is
    possible that the biker simply ignored the questions and did
    not respond to both 'like' and 'dislike' entries, hence values
    maybe 0,0 for both last columns.

tour_convoy.csv
tour_convoy.csv consists the list of bikers that showed interest
    in a particular tour.
tour_id: Unique identifier for particular tour.
going: Space-delimited list of bikers who said they will go to the
     tour.
maybe: Space-delimited list of bikers who said they might go to
    the tour.
invited: Space-delimited list of bikers who were invited to the
    tour.
not_going: Space-delimited list of bikers who said they will not
    go to the tour.

bikers.csv
bikers.csv has feature information about bikers.
biker_id: Unique identifier for a biker person.
language_id: Identifier of the language biker speaks.
location_id: Identifier of the location biker resides in.
bornIn: Year of birth of the biker to estimate their age.
gender: male/female based on their bikers input.
member_since: Date of joining the bikers interest group.
area: bikers location (if known).
time_zone: this is the offset in minutes to GMT timezone. (For
    example Indian Time is +5:30 GMT, so +330 minutes).

tours.csv
```

```
tours.csv consists feature information about the tours.
tour_id: Unique identifier for particular tour.
biker_id: ID of the biker who organized the tour.
tour_date: date on which tour was conducted.
city: location of tour (if known)
state: location of tour (if known)
pincode: location of tour (if known)
country: location of tour (if known)
latitude: approximate location of the starting point of the tour (
    if known)
longitude: approximate location of the starting point of the tour
    (if known)
w1, w2,    , w100: Number of occurences of most common words in
    the description of the tour. We took 100 most common/important
    words among all the descriptions provided in the tour guide,
    and each column w1, w2,    w100 gives the count of number of
    times each word w_i has occured in the description of a given
    tour_id.
w_other: count of other words.

bikers_network.csv
bikers_network.csv consists of the social networks of the bikers.
    This is derived from the group of bikers that are know each
    other via some groups.
biker_id: unique id for a biker
friends: this is a list of all friends of given biker_id (Note:
    this is a space delimited column).

PLEASE NOTE
We don't have data of invitees/likes of all tours in tours.csv and
    tour_convoy.csv, and hence any given biker/tour in tours.csv
    or tour_convoy.csv may not appear in train.csv.

SUBMISSION-SPECIFIC INSTRUCTIONS
------

Your submission consists of the implemented code only. Ensure that
    all required functions are present in 'submission' with the
    correct signatures at 'submission/'.
We will only evaluate submission found at this path.

- You MUST implement the following Python functions with the exact
    signatures. Changing their names, arguments, or return types
    will result in disqualification.
- You may define additional functions, classes, or modules as
    needed to implement your solution, as long as the required
    functions above are present with the correct signatures.
- Your submission must be contained in a **single Python file**.
    All code, including any helper functions or classes, should
    reside in this file. The evaluation will only consider this
    file as the final solution.
- The 'run()' function is PROVIDED AS-IS and MUST NOT be modified
    in any way.

**CRITICAL CONSTRAINTS:**
- **NO GLOBAL VARIABLES**: Do not define any variables outside of
    function/class scope. All data must be contained within
    functions or class methods.
- **IMPORT RESTRICTION**: During evaluation, only the functions
    will be imported and called. Any code outside function/class
    definitions (except imports and class definitions) will be
    IGNORED and may cause errors.
```

```python
- **SELF-CONTAINED**: All necessary initialization code must be
    placed inside the functions that need it. Do not rely on global
     state.
```

Listing 2: Mono predict & extended arguments variant

```python
'''python
import pandas as pd
import numpy as np
from typing import TypedDict

def train_and_predict(bikers: pd.DataFrame, tours: pd.DataFrame,
    tour_convoy: pd.DataFrame, bikers_network: pd.DataFrame, data:
    pd.DataFrame, bikers_val: pd.DataFrame, tours_val: pd.DataFrame
    , tour_convoy_val: pd.DataFrame, bikers_network_val: pd.
    DataFrame, data_val: pd.DataFrame) -> np.ndarray:
     """
     This function takes the training data, validation features and
         returns the predictions for validation features

     Args:
         bikers (pd.DataFrame): Biker demographic information
         tours (pd.DataFrame): Tour features and word counts
         tour_convoy (pd.DataFrame): Tour participation lists
         bikers_network (pd.DataFrame): Social network connections
         data (pd.DataFrame): Train data
         bikers_val (pd.DataFrame): Biker demographic information
         tours_val (pd.DataFrame): Tour features and word counts
         tour_convoy_val (pd.DataFrame): Tour participation lists
         bikers_network_val (pd.DataFrame): Social network
             connections
         data_val (pd.DataFrame): Validation data
     """
     ...
''
```

Listing 3: Mono predict & short arguments variant

```python
'''python
import pandas as pd
import numpy as np
from typing import TypedDict

class X_trainDict(TypedDict):
    bikers: pd.DataFrame
    tours: pd.DataFrame
    tour_convoy: pd.DataFrame
    bikers_network: pd.DataFrame
    data: pd.DataFrame

class X_valDict(TypedDict):
    bikers: pd.DataFrame
    tours: pd.DataFrame
    tour_convoy: pd.DataFrame
    bikers_network: pd.DataFrame
    data: pd.DataFrame

def train_and_predict(X_train: X_trainDict, X_val: X_valDict) ->
    np.ndarray:
```

```
        """
        This function takes the training data, validation features and
            returns the predictions for validation features

        Args:
            X_train (dict[str, pd.DataFrame]): dict with the following
                keys.
                Expected keys:
                bikers (pd.DataFrame): Biker demographic information
                tours (pd.DataFrame): Tour features and word counts
                tour_convoy (pd.DataFrame): Tour participation lists
                bikers_network (pd.DataFrame): Social network
                    connections
                data (pd.DataFrame): Train data
            X_val (dict[str, pd.DataFrame]): dict with the following
                keys.
                Expected keys:
                bikers (pd.DataFrame): Biker demographic information
                tours (pd.DataFrame): Tour features and word counts
                tour_convoy (pd.DataFrame): Tour participation lists
                bikers_network (pd.DataFrame): Social network
                    connections
                data (pd.DataFrame): Validation data
        """
        ...
```
```

Listing 4: Modular predict & short arguments variant

```python
import pandas as pd
import numpy as np
from typing import Any, TypedDict

class X_trainDict(TypedDict):
    bikers: pd.DataFrame
    tours: pd.DataFrame
    tour_convoy: pd.DataFrame
    bikers_network: pd.DataFrame
    data: pd.DataFrame

class X_valDict(TypedDict):
    bikers: pd.DataFrame
    tours: pd.DataFrame
    tour_convoy: pd.DataFrame
    bikers_network: pd.DataFrame
    data: pd.DataFrame

def train(X_train: X_trainDict) -> Any:
    """
    This function takes training data and returns the trained
        model and any intermediate variables

    Args:
        X_train (dict[str, pd.DataFrame]): dict with the following
            keys.
            Expected keys:
            bikers (pd.DataFrame): Biker demographic information
            tours (pd.DataFrame): Tour features and word counts
            tour_convoy (pd.DataFrame): Tour participation lists
```

```
                            bikers_network (pd.DataFrame): Social network
                                connections
                            data (pd.DataFrame): Train data
        """
        ...

    def prepare_val(train_output: Any, X_val: X_valDict) -> Any:
        """
        This function takes train function output and processed
            validation features

        Args:
            X_val (dict[str, pd.DataFrame]): dict with the following
                keys.
                Expected keys:
                bikers (pd.DataFrame): Biker demographic information
                tours (pd.DataFrame): Tour features and word counts
                tour_convoy (pd.DataFrame): Tour participation lists
                bikers_network (pd.DataFrame): Social network
                    connections
                data (pd.DataFrame): Validation data
            train_output (Any): Output from the train function
        """
        ...

    def predict(train_output: Any, prepare_val_output: Any) -> np.
        ndarray:
        """
        This function takes train and prepare_val functions outputs
            and generates the prediction for validation features

        Args:
            train_output (Any): Output from the train function
            prepare_val_output (Any): Output from the prepare_val
                function, which is the processed X_val dataframe
        """
        ...

    def run(X_train: X_trainDict, X_val: X_valDict) -> np.ndarray:
        """
        This function takes the training data, validation features and
             returns the predictions for validation features

        Args:
            X_train (dict[str, pd.DataFrame]): dict with the following
                 keys.
                Expected keys:
                bikers (pd.DataFrame): Biker demographic information
                tours (pd.DataFrame): Tour features and word counts
                tour_convoy (pd.DataFrame): Tour participation lists
                bikers_network (pd.DataFrame): Social network
                    connections
                data (pd.DataFrame): Train data
            X_val (dict[str, pd.DataFrame]): dict with the following
                keys.
                Expected keys:
                bikers (pd.DataFrame): Biker demographic information
                tours (pd.DataFrame): Tour features and word counts
                tour_convoy (pd.DataFrame): Tour participation lists
                bikers_network (pd.DataFrame): Social network
                    connections
                data (pd.DataFrame): Validation data
```

```
1026          """
1027          train_output = train(X_train)
1028          return predict(train_output, prepare_val(train_output, X_val))
1029      ```
```

Listing 5: Modular predict & extended arguments variant

```python
1033  ```python
1034  import pandas as pd
1035  import numpy as np
1036  from typing import Any, TypedDict
1037
1038
1039  def train(bikers: pd.DataFrame, tours: pd.DataFrame, tour_convoy:
1040      pd.DataFrame, bikers_network: pd.DataFrame, data: pd.DataFrame)
1041       -> Any:
1042      """
1043      This function takes training data and returns the trained
1044          model and any intermediate variables
1045
1046      Args:
1047          bikers (pd.DataFrame): Biker demographic information
1048          tours (pd.DataFrame): Tour features and word counts
1049          tour_convoy (pd.DataFrame): Tour participation lists
1050          bikers_network (pd.DataFrame): Social network connections
1051          data (pd.DataFrame): Train data
1052      """
1053      ...
1054
1055
1056  def prepare_val(train_output: Any, bikers_val: pd.DataFrame,
1057      tours_val: pd.DataFrame, tour_convoy_val: pd.DataFrame,
1058      bikers_network_val: pd.DataFrame, data_val: pd.DataFrame) ->
1059      Any:
1060      """
1061      This function takes train function output and processed
1062          validation features
1063
1064      Args:
1065          bikers_val (pd.DataFrame): Biker demographic information
1066          tours_val (pd.DataFrame): Tour features and word counts
1067          tour_convoy_val (pd.DataFrame): Tour participation lists
1068          bikers_network_val (pd.DataFrame): Social network
1069              connections
1070          data_val (pd.DataFrame): Validation data
1071          train_output (Any): Output from the train function
1072      """
1073      ...
1074
1075
1076  def predict(train_output: Any, prepare_val_output: Any) -> np.
1077      ndarray:
1078      """
1079      This function takes train and prepare_val functions outputs
          and generates the prediction for validation features

      Args:
          train_output (Any): Output from the train function
          prepare_val_output (Any): Output from the prepare_val
              function, which is the processed X_val dataframe
      """
      ...
```

```
def run(bikers: pd.DataFrame, tours: pd.DataFrame, tour_convoy: pd
    .DataFrame, bikers_network: pd.DataFrame, data: pd.DataFrame,
    bikers_val: pd.DataFrame, tours_val: pd.DataFrame,
    tour_convoy_val: pd.DataFrame, bikers_network_val: pd.DataFrame
    , data_val: pd.DataFrame) -> np.ndarray:
    """
    This function takes the training data, validation features and
        returns the predictions for validation features

    Args:
        bikers (pd.DataFrame): Biker demographic information
        tours (pd.DataFrame): Tour features and word counts
        tour_convoy (pd.DataFrame): Tour participation lists
        bikers_network (pd.DataFrame): Social network connections
        data (pd.DataFrame): Train data
        bikers_val (pd.DataFrame): Biker demographic information
        tours_val (pd.DataFrame): Tour features and word counts
        tour_convoy_val (pd.DataFrame): Tour participation lists
        bikers_network_val (pd.DataFrame): Social network
            connections
        data_val (pd.DataFrame): Validation data
    """
    train_output = train(bikers, tours, tour_convoy,
        bikers_network, data)
    return predict(train_output, prepare_val(train_output,
        bikers_val, tours_val, tour_convoy_val, bikers_network_val,
        data_val))
```
```

## E  TRANSLATION PROMPT

Since some fields include imperatives (e.g., *Develop a model*, *Create an agent*), it has to be defined explicitly in a prompt (Figure 6) to use imperative mood, otherwise the model have translated English imperatives, which have the same form as verbs not in imperative mood, mainly as infinitives.

Prompt example

Translate text into **{target_language}**. Infinitive forms that stand apart, if any, should be translated as the imperative mood: **{text}**

Figure 6: Example of the prompt used in translation experiments.

## F  FORM EXAMPLE

In Figure 7 there is an example of one question block in a form, which asks a native speaker of Romanian to validate the translation of the competition description..

## G  VALIDATION RESULTS

As it has been stated before, for each language we considered the translations from a single annotator. To prove the reliability and the quality of translations, we have used back-translation into English and have assessed quality using BLEU metric.

Using the same GPT-4o model and identical prompts, we have executed three independent translation runs and collected the resulting English back-translations. We then have computed BLEU scores between the original texts in English and their back-translated versions. Finally, we have estimated bootstrap confidence intervals for each language and text type.

**Translated version:**

Dezvoltați un model predictiv pentru a anticipa probabilitatea mortalității în spital pentru pacienți. Seturile de date includ diverse caracteristici legate de pacienți la momentul internării în spital. Obiectivul este de a prezice cu acuratețe probabilitatea mortalității în spital pentru fiecare pacient din setul de testare.

**Original version:**

Develop a predictive model to forecast the likelihood of hospital mortality for patients. The datasets include various features related to the patients upon hospital admission. The objective is to predict the probability of hospital mortality for each patient in the test set accurately.

**Does the translated text (1) sound native and (2) convey the same meaning as the original text?**

- YES and YES

- NO and YES

- YES and NO

- NO and NO

If there is at least one NO in the answer, please suggest your own version:

Figure 7: Example of question block in Google Form for Romanian language

Lavie (2010) claims that BLEU score in range [0.4, 0.5] can be a sign of high-quality translation. As shown in Figure 8, the mean BLEU scores across the three runs exceed 0.4 for all languages and text types except Kazakh data-description texts. Moreover, the intervals for descriptions and data cards are relatively narrow, suggesting stable and accurate translations.

For domain-level texts, the confidence intervals are notably wider. This is expected since BLEU is highly sensitive to short texts, and GPT-4o often produces less consistent translations for very short inputs lacking contextual cues. Overall, these results support our claim that translations from a single native speaker are sufficiently reliable for our study.

## H   GPT-4 TRANSLATION PATTERNS

Prior work shows that GPT-4 produces more accurate and more lexically diverse translations than commercial systems such as Google Translate (Jiao et al., 2023). Additionally, Raunak et al. (2023) note that GPT-family models tend to generate non-literal translations, including figurative renderings of idioms. Based on this, we anticipated many outputs that preserved meaning but lacked features that make them sound fully native.

Figure 9 indicates that nearly two-thirds of responses within each language are both judged natural and semantically equivalent, with Romanian and Kazakh exhibiting the lowest proportions among the evaluated languages. This pattern is consistent with evidence that GPT-based translation quality degrades for low-resource languages, which typically have fewer native speakers and, thus, less training data available (Hendy et al., 2023).

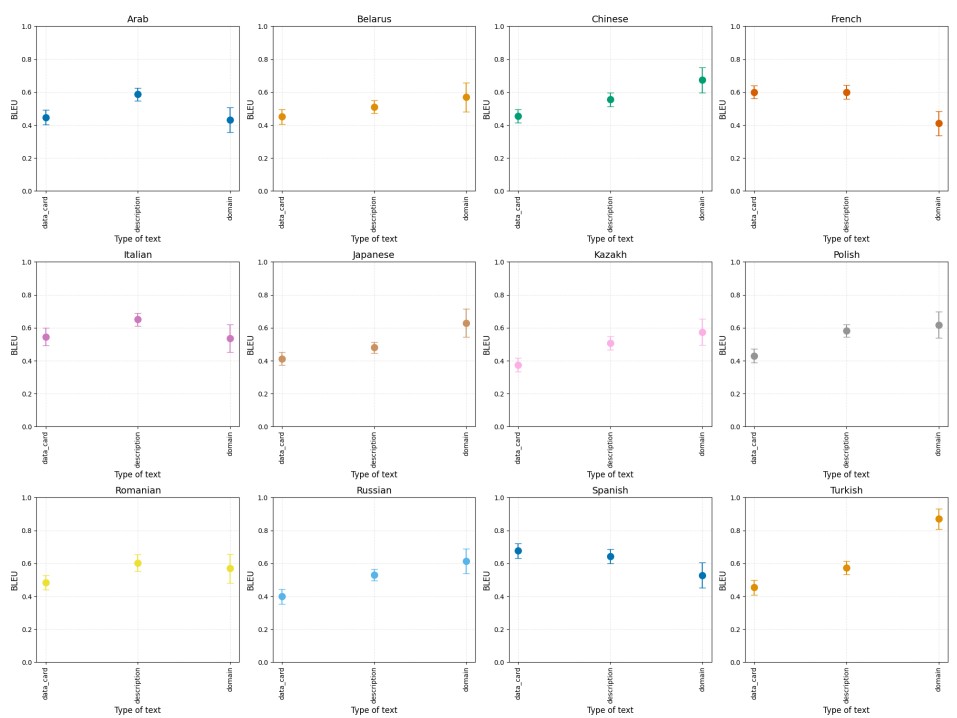

Figure 8: Confidence intervals for each language and type of text

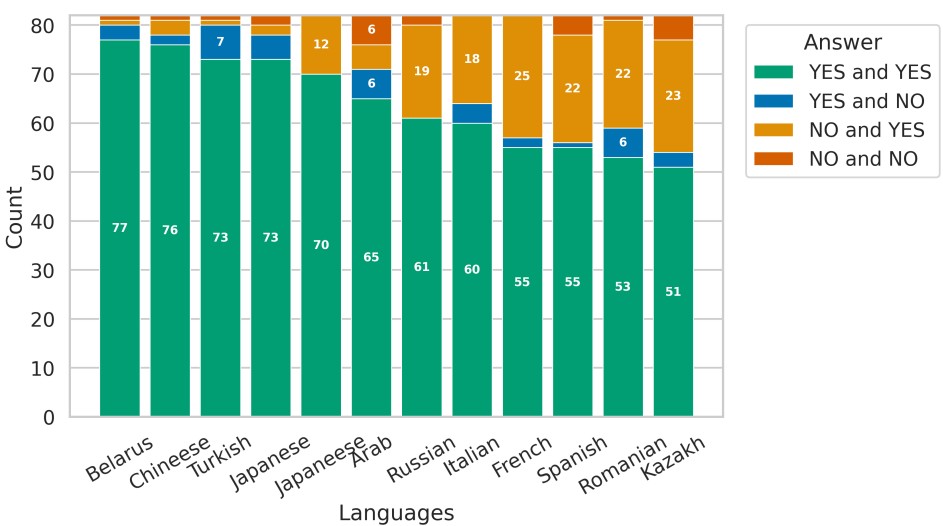

Figure 9: Distribution of response types within each language

Figure 10 (A) further shows that GPT systems predominantly produce natural translations without semantic distortion, with only 1.3 % of all outputs rated neither natural nor semantically similar to the source and with almost 4 % of translations sounding natural but conveying other meaning. The share of labeled "NO and YES" suggests that, while models preserve meaning, they often employ more varied and less concise phrasings in the target language.

# I  CODE EXAMPLES

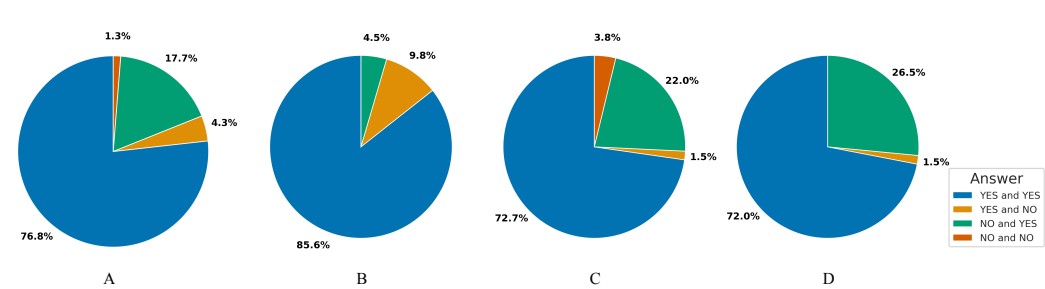

Figure 10: Distribution of translation evaluation outcomes. (A) Overall distribution across all languages; (B–D) distributions for the evaluation of translations of domains (B), data cards (C), and task descriptions (D), respectively.

Listing 6: Valid code example

```python
import pandas as pd
import numpy as np
from typing import Any
from sklearn.model_selection import GroupKFold
from sklearn.metrics import mean_absolute_error
import lightgbm as lgb

def create_features(df):
    # Basic features
    df["R"] = df["R"].astype("category")
    df["C"] = df["C"].astype("category")

    # Lag features for u_in and u_out to capture time series
    df["u_in_lag1"] = df.groupby("breath_id")["u_in"].shift(1).
        fillna(0)
    df["u_in_lag2"] = df.groupby("breath_id")["u_in"].shift(2).
        fillna(0)
    df["u_out_lag1"] = df.groupby("breath_id")["u_out"].shift(1).
        fillna(0)
    df["u_out_lag2"] = df.groupby("breath_id")["u_out"].shift(2).
        fillna(0)

    # Time step as feature
    df["time_step"] = df["time_step"].astype(np.float32)

    # We can also add cumulative sum of u_in per breath to capture
        progression
    df["u_in_cumsum"] = df.groupby("breath_id")["u_in"].cumsum()

    # Drop columns not used for training
    return df

def train(X_train: pd.DataFrame, y_train: pd.DataFrame) -> Any:
    X_train = create_features(X_train)
    features = [
        "R",
        "C",
        "time_step",
        "u_in",
        "u_out",
        "u_in_lag1",
```

```
39            "u_in_lag2",
40            "u_out_lag1",
41            "u_out_lag2",
42            "u_in_cumsum",
43        ]
44        categorical_features = ["R", "C", "u_out", "u_out_lag1", "
              u_out_lag2"]
45
46        # Prepare dataset for LightGBM
47        lgb_train = lgb.Dataset(
48            X_train[features],
49            label=y_train["pressure"],
50            categorical_feature=categorical_features,
51            free_raw_data=False,
52        )
53
54        params = {
55            "objective": "regression",
56            "metric": "mae",
57            "boosting_type": "gbdt",
58            "learning_rate": 0.1,
59            "num_leaves": 31,
60            "max_depth": -1,
61            "verbose": -1,
62            "seed": 42,
63            "feature_pre_filter": False,
64        }
65
66        model = lgb.train(params, lgb_train, num_boost_round=100)
67
68        return (model, features, categorical_features)
69
70
71  def prepare_val(X_val: pd.DataFrame, train_output: Any) -> Any:
72        model, features, categorical_features = train_output
73        X_val = create_features(X_val)
74        return X_val[features]
75
76
77  def predict(train_output: Any, prepare_val_output: Any) -> np.
          array:
78        model, features, categorical_features = train_output
79        X_val = prepare_val_output
80        preds = model.predict(X_val, num_iteration=model.
              best_iteration)
81        return preds
82
83
84  def run(X_train: pd.DataFrame, y_train: pd.DataFrame, X_val: pd.
          DataFrame) -> np.array:
85        train_output = train(X_train, y_train)
86        X_val_processed = prepare_val(X_val, train_output)
87        preds = predict(train_output, X_val_processed)
88        return preds
```

Listing 7: Crashed due to unsupported lgb.train parameter (line 65)

```
1  import pandas as pd
2  import numpy as np
3  from typing import Any
4  from sklearn.model_selection import GroupKFold
5  from sklearn.metrics import mean_absolute_error
6  import lightgbm as lgb
7  import os
8
```

```python
9
10  def feature_engineering(df: pd.DataFrame) -> pd.DataFrame:
11      # Basic features
12      df["u_in_cumsum"] = df.groupby("breath_id")["u_in"].cumsum()
13      df["u_in_lag1"] = df.groupby("breath_id")["u_in"].shift(1).
            fillna(0)
14      df["u_in_lag2"] = df.groupby("breath_id")["u_in"].shift(2).
            fillna(0)
15      df["u_out_lag1"] = df.groupby("breath_id")["u_out"].shift(1).
            fillna(0)
16      df["time_step_diff"] = df.groupby("breath_id")["time_step"].
            diff().fillna(0)
17      # Interaction features
18      df["R*C"] = df["R"] * df["C"]
19      df["R*u_in"] = df["R"] * df["u_in"]
20      df["C*u_in"] = df["C"] * df["u_in"]
21      return df
22
23
24  def train(X_train: pd.DataFrame, y_train: pd.DataFrame) -> Any:
25      X_train = feature_engineering(X_train)
26      features = [
27          "R",
28          "C",
29          "time_step",
30          "u_in",
31          "u_out",
32          "u_in_cumsum",
33          "u_in_lag1",
34          "u_in_lag2",
35          "u_out_lag1",
36          "time_step_diff",
37          "R*C",
38          "R*u_in",
39          "C*u_in",
40      ]
41      X_train = X_train[features]
42      y = y_train["pressure"].values
43
44      # GroupKFold on breath_id to avoid leakage
45      # We will train a single model on full data here for
            simplicity
46      params = {
47          "objective": "regression",
48          "metric": "mae",
49          "boosting_type": "gbdt",
50          "learning_rate": 0.1,
51          "num_leaves": 64,
52          "feature_fraction": 0.8,
53          "bagging_fraction": 0.8,
54          "bagging_freq": 5,
55          "verbose": -1,
56          "seed": 42,
57          "n_jobs": -1,
58      }
59      lgb_train = lgb.Dataset(X_train, y)
60      model = lgb.train(
61          params,
62          lgb_train,
63          num_boost_round=1000,
64          valid_sets=[lgb_train],
65          early_stopping_rounds=50, # <-- Unsupported since version
                4.0.0
66          verbose_eval=100,
67      )
```

```
68        return (model, features)
69
70
71  def prepare_val(X_val: pd.DataFrame, train_output: Any) -> Any:
72      model, features = train_output
73      X_val = feature_engineering(X_val)
74      X_val = X_val[features]
75      return X_val
76
77
78  def predict(train_output: Any, prepare_val_output: Any) -> np.
        array:
79      model, _ = train_output
80      X_val = prepare_val_output
81      preds = model.predict(X_val, num_iteration=model.
            best_iteration)
82      return preds
83
84
85  def run(X_train: pd.DataFrame, y_train: pd.DataFrame, X_val: pd.
        DataFrame) -> np.array:
86      train_output = train(X_train, y_train)
87      prepared_val = prepare_val(X_val, train_output)
88      preds = predict(train_output, prepared_val)
89      return preds
```

Listing 8: Using target column 'presure' in validation data preprocessing (lines 14-15)

```
1   import pandas as pd
2   import numpy as np
3   from typing import Any, Tuple
4   from sklearn.model_selection import train_test_split
5   from sklearn.metrics import mean_absolute_error
6   import lightgbm as lgb
7
8   def add_features(df: pd.DataFrame) -> pd.DataFrame: # <-- using
        target column presure
9       # Sort by breath_id and time_step for lag features
10      df = df.sort_values(["breath_id", "time_step"]).reset_index(
            drop=True)
11      # Lag features for u_in and pressure
12      df["u_in_lag1"] = df.groupby("breath_id")["u_in"].shift(1).
            fillna(0)
13      df["u_in_lag2"] = df.groupby("breath_id")["u_in"].shift(2).
            fillna(0)
14      df["pressure_lag1"] = df.groupby("breath_id")["pressure"].
            shift(1).fillna(0)
15      df["pressure_lag2"] = df.groupby("breath_id")["pressure"].
            shift(2).fillna(0)
16      # Rolling mean of u_in over last 3 steps
17      df["u_in_roll3"] = (
18          df.groupby("breath_id")["u_in"]
19          .rolling(3, min_periods=1)
20          .mean()
21          .reset_index(0, drop=True)
22      )
23      # Rolling std dev of u_in over last 3 steps
24      df["u_in_roll3_std"] = (
25          df.groupby("breath_id")["u_in"]
26          .rolling(3, min_periods=1)
27          .std()
28          .reset_index(0, drop=True)
29          .fillna(0)
30      )
31      # Time step squared to capture nonlinear trend
```

```
32        df["time_step2"] = df["time_step"] ** 2
33        return df
34
35
36  def train(X_train: pd.DataFrame, y_train: pd.DataFrame) -> Any:
37      # Add features to training data
38      X_train = add_features(pd.concat([X_train, y_train], axis=1))
39      # Target
40      y = X_train["pressure"].values
41      # Drop columns not used as features
42      drop_cols = ["id", "breath_id", "pressure"]
43      features = [c for c in X_train.columns if c not in drop_cols]
44      X = X_train[features]
45      # LightGBM dataset
46      lgb_train = lgb.Dataset(X, label=y)
47      params = {
48          "objective": "regression",
49          "metric": "mae",
50          "verbosity": -1,
51          "boosting_type": "gbdt",
52          "learning_rate": 0.1,
53          "num_leaves": 31,
54          "feature_fraction": 0.9,
55          "bagging_fraction": 0.9,
56          "bagging_freq": 1,
57          "seed": 42,
58          "n_jobs": -1,
59      }
60      model = lgb.train(params, lgb_train, num_boost_round=300)
61      return (model, features)
62
63
64  def prepare_val(X_val: pd.DataFrame, train_output: Any) -> Any:
65      model, features = train_output
66      # Add features to validation data (pressure not available,
              fill with 0)
67      X_val = add_features(X_val.copy())
68      # For lags of pressure which we don't have in val, fill with 0
69      X_val["pressure"] = 0.0
70      X_val["pressure_lag1"] = 0.0
71      X_val["pressure_lag2"] = 0.0
72      X_val = add_features(X_val)
73      return X_val[features]
74
75
76  def predict(train_output: Any, prepare_val_output: Any) -> np.
          array:
77      model, features = train_output
78      X_val = prepare_val_output
79      preds = model.predict(X_val)
80      return preds
81
82
83  def run(X_train: pd.DataFrame, y_train: pd.DataFrame, X_val: pd.
          DataFrame) -> np.array:
84      train_output = train(X_train, y_train)
85      val_features = prepare_val(X_val, train_output)
86      preds = predict(train_output, val_features)
87      return preds
```

