# OpenReview forum: "ML²B: Multi-Lingual ML Benchmark For AutoML"
_ICLR.cc/2026/Conference — ICLR 2026 Conference Withdrawn Submission_

### Official Review · Reviewer_pm4b · 2025-10-17

**Soundness:** 3
**Presentation:** 2
**Contribution:** 1
**Rating:** 2
**Confidence:** 4

**Summary:**

This paper introduces ML2B, a benchmark designed to evaluate Large Language Models (LLMs) on multilingual machine learning code generation. ML2B extends 30 Kaggle competitions by translating them into 13 languages, covering tabular, text, and image data modalities. The benchmark incorporates human-validated translations, structured metadata for prompts, and leakage-prevention techniques such as private test sets and modular code formats. Through the AIDE evaluation framework, the authors assess multiple LLMs and report a consistent performance degradation of 15–45% on non-English prompts compared to English, with the drop being particularly pronounced for low-resource languages.

**Strengths:**

1. Uses real-world Kaggle competitions.
2. The translation is verified by human reviewers.
3. The use of percentile-based leaderboard scoring is an appropriate and effective choice for normalizing performance and enabling fair comparisons across heterogeneous competitions.

**Weaknesses:**

1. The translation validation process, while including human oversight, lacks quantitative quality measures such as inter-annotator agreement (IAA) scores. This makes it difficult to objectively assess the quality and consistency of the translations.
2. The discussion of the results is largely descriptive, reporting what performance drops occurred, but fails to provide a deep, explanatory analysis of why they occurred.
3.  While the creation of a new benchmark is a contribution, the novelty of this specific benchmark feels limited. More importantly, the paper lacks substantive insights derived from the results. It does not offer in-depth analysis of model failure modes or propose concrete strategies to mitigate the observed performance gap in multilingual settings.
4. The overall presentation could be improved.  Additionally, some figures are difficult to interpret due to low resolution or unclear labeling, which hinders readability.

**Questions:**

1. Have the authors considered evaluating more recent or capable models (e.g., GPT-5, Gemini 2.5 Pro)? Evaluating state-of-the-art models would provide a more current snapshot of LLM capabilities on this benchmark.
2. A simple yet crucial baseline seems to be missing: what is the performance if the LLM is explicitly prompted to first translate the non-English problem description into English and then generate the solution?
3. The superscripts (¹ and ²) used in Table 1 are not defined in the table's caption or the main body of the paper. Could you please clarify their meaning?

**Details Of Ethics Concerns:**

The authors state that human annotators who performed the crucial task of validating translations received "no monetary compensation" (Appendix F). Relying on significant unpaid labor for resource creation raises ethical concerns regarding fair treatment and compensation, which may conflict with the principles outlined in the ICLR Code of Ethics.

---

> ### Author Response · Authors · 2025-12-04
>
> Thank you for these valuable suggestions. Evaluating more recent frontier models such as GPT-5 or Gemini 2.5 Pro is indeed an important extension; at the time of our experiments, these models were unavailable. We also appreciate the idea of adding a “translate to English → generate code” baseline, including structured prompting variants such as OH and ReAct, as this will help isolate the contribution of translation quality to downstream performance; we will incorporate this baseline in future iterations.
>
> >The translation validation process, while including human oversight, lacks quantitative quality measures such as inter-annotator agreement (IAA) scores.
>
> We included a detailed analysis of BLEU scores. The mean BLEU scores across three runs exceed 0.4 (typically considered a satisfactory level) for all languages and text types except Kazakh data-description texts. Moreover, the intervals for descriptions and data cards are relatively narrow, suggesting stable and accurate translations.
>
> For domain-level texts, the confidence intervals are notably wider. This is expected, as BLEU is highly sensitive to short texts, and GPT-4o may produce less consistent translations for brief inputs lacking contextual cues. Detailed confidence-interval plots are included in the revised paper.
>
> >A simple yet crucial baseline seems to be missing: what is the performance if the LLM is explicitly prompted to first translate the non-English problem description into English and then generate the solution?
>
> This is a valid insight, and this metric is going to be added in the future. We couldn't incorporate this baseline in the current revision due to the essential requirement of finding native English speakers.
>
> > The superscripts (¹ and ²) used in Table 1 are not defined in the table's caption or the main body of the paper. Could you please clarify their meaning?
>
> Thank you for the comment, we have revised the format of the tables

---

### Official Review · Reviewer_RbCc · 2025-10-31

**Soundness:** 1
**Presentation:** 1
**Contribution:** 2
**Rating:** 2
**Confidence:** 4

**Summary:**

The paper introduces ML²B, the first benchmark for evaluating multilingual ML code generation by LLMs. It expands 30 Kaggle competitions into 13 languages with validated translations, metadata, and leakage-controlled setups. Evaluation across AIDE and ML Master on 5 LLMs shows a 15–45% drop in performance on non-English tasks, especially for low-resource languages like Kazakh and Belarusian.

**Strengths:**

- This paper is the first multilingual benchmark targeting end-to-end ML pipeline generation, filling a clear gap beyond English-only datasets like MLE-Bench or DA-Code.
- Careful translation-quality assessment across 13 languages with native speakers.
- The discovery of performance degradation on non-English tasks is interesting.

**Weaknesses:**

- The benchmarked frameworks are limited to AIDE and ML Master. The paper omits key baselines, including commercial tools such as Claude Code, Codex, and Cursor, and open-source alternatives such as OpenHands, SWE-agents, and Aider. As a comprehensive benchmark, it would be interesting to see how these major frameworks perform. Moreover, a human baseline is always good to have.

- It seems the paper borrowed a lot of concepts and content from the AIDE paper. The benchmark for Kaggle competitions has been explored extensively by previous works (e.g., MLE-Bench, AutoKaggle). This undermines the paper's novelty.

- The result section is hard to follow; none of these error analysis examples are provided. No conclusions are drawn from that.

- The authors should at least provide some of the generated code examples to show the quality. If possible, a leaderboard figure would be beneficial.

- The pie charts are poorly made and meaningless. For example, lower values are better for loss metrics, while higher values are better for accuracy metrics. How can you compare them in the same chart?

- The authors should pay attention to the format of the tables. For example, Table 4 has redundant numbers, and Table 5 falls off the page.

**Questions:**

- (Just for curiosity) Native speakers’ varying computer science skills may cause bias in translated benchmark tasks across languages. A sanity check—translating tasks into a target language and back into English with an LLM, then re-evaluating—can validate translation quality. Significant performance drops suggest semantic drift in that language version.

---

> ### Author Response · Authors · 2025-12-04
>
> We sincerely thank the reviewer for their valuable comments and for the important suggestion to use back-translation to assess translation quality. Due to time constraints, we were unable to recruit a sufficient number of native English speakers to manually evaluate the back-translations. Instead, we computed BLEU scores between the original texts and their back-translations and derived corresponding confidence intervals. These analyses indicate that the translation quality is consistently high, supporting the reliability of our single–native-speaker validation setup. Further details on our translation validation procedure are provided in the Validating translations section of the paper.
>
> > It seems the paper borrowed a lot of concepts and content from the AIDE paper. The benchmark for Kaggle competitions has been explored extensively by previous works (e.g., MLE-Bench, AutoKaggle). This undermines the paper's novelty.
>
> We appreciate the reviewer’s observation regarding prior work on Kaggle-based benchmarks, such as MLE-Bench, AutoKaggle, and AIDE. While these benchmarks have indeed explored various aspects of ML code generation, none of them incorporate the combination of features that ML²B introduces, nor do they target the multilingual and leakage-robust setting that is the focus of our work. Existing benchmarks operate exclusively in English, do not include private competitions to mitigate training-data leakage, and do not evaluate complete ML pipelines under isolated execution environments. In contrast, ML²B is the first benchmark to unify multilingual problem descriptions, leakage-resistant task selection, and secure, isolated code execution
>
> > The authors should at least provide some of the generated code examples to show the quality.
>
> Thank you for this vital suggestion. We have added code examples into Appendix part.

---

### Official Review · Reviewer_QtnP · 2025-11-01

**Soundness:** 2
**Presentation:** 2
**Contribution:** 2
**Rating:** 2
**Confidence:** 3

**Summary:**

This paper proposes a Kaggle-grounded benchmark for evaluting the AutoML code generation capability of LLMs (and agents). Importantly, the authors selected the seed English problems and translated into multiple (major and minor) languages to assess the performance when presented with non-English problem context.
The performance is evaluated as the score percentile in the leaderboard, which allows for a consistent measure across various Kaggle tasks. Mixed results across languages suggest data leakage.

**Strengths:**

* Important contribution to expand the coverage of the tasks to non-English languages
* Comprehensive descriptions on the dataset construction process

**Weaknesses:**

* Paper organization needs improvement.
  * Majority of the paper discuss the data construction, which I agree is important to clarify and be transparent, but the findings and experimental resuluts should be expanded even more in the main body.
  * One of the main results (Table 4) is mentioned in the main text, but appears in the Appendix. The authors should organize tables and figures within the main text (9 pages) if they are discussed primarily in the main text.
* I might be misinterpreting, but Table 4 doesn't seem to indicate that English consistently yields lower percentiles than other languages. I'd be happy to be corrected about this.
* The motivation and overall process are sound to me. For the paper to be more appropriate for ICLR, I would suggest the content in the main text to focus more on the performance analysis in cross-lingual settings instead of the Related work and the data curation process. I commend the authors for the level of details in both related work and the data construction -- I just think some portion could be compressed or be included in the appendix.

**Questions:**

* Table 4 has strange "41.8cm" and "31.8cm" before the model names. Perhaps it's incorrectly escaped LaTeX.

---

> ### Author Response · Authors · 2025-12-04
>
> We are grateful for the reviewer's valuable insights and advice. We have reduced the data construction part, transferred some points to appendix giving more space for results discussion.
>
> > Table 4 doesn't seem to indicate that English consistently yields lower percentiles than other languages
>
> In the updated submission, we replaced percentile-based rankings with a more stable and comparable metric: AUP (Area Under the Performance Profile). After recomputing results with AUP we have come to the following conclusions:
> * Languages with rich morphology and limited representation in pretraining data (e.g., Arabic, Belarusian) exhibit consistently low performance, likely due to lexical sparsity and structural complexity.
> * Languages with non-Latin scripts but substantial pretraining presence (e.g., Japanese) show highly unstable performance, suggesting that tokenization issues and script‑specific encoding drive their inconsistency.

---

### Official Review · Reviewer_Ader · 2025-11-01

**Soundness:** 2
**Presentation:** 2
**Contribution:** 3
**Rating:** 2
**Confidence:** 4

**Summary:**

ML²B is a new benchmark for multilingual ML code generation: the authors curate 30 Kaggle competitions (24 public + 6 private), translate the task descriptions / data cards into 13 natural languages, and build an end-to-end evaluation pipeline (BenchPipeline + Docker runtime + AST-based grader). They evaluate five LLMs across two execution frameworks (AIDE and ML-Master hybrids), report systematic performance drops (claimed 15–45%) on non-English prompts, show domain-dependent variability, and run static leakage analysis on generated code. The benchmark includes structured metadata and human review of translations.

**Strengths:**

1. The motivation is strong and timely. The paper identifies a real blind spot: existing ML code-generation benchmarks are English-only while real users and problem descriptions could be multilingual.

2. Using Kaggle competitions and executing generated code (via Docker / AST transformations and a grader) is closer to practical utility than small snippet tests.

3. Covering 13 natural languages (including low-resource ones like Kazakh, Belarusian, Romanian) is valuable and exposes cross-lingual weaknesses that English-centric benchmarks hide.

**Weaknesses:**

1. Small/possibly biased task set where 30 competitions is useful but relatively small. It’s unclear how representative the selected 24 public + 6 private competitions are of Kaggle’s diversity. The selection process relies on LLM filters and manual review which can introduce bias but details are sparse (Appendix D doesn't give a clear picture on how the diversity in task selection is ensured).

2. Translations were generated with GPT-4o and validated by a single native-speaker annotator per text (no compensation). The authors state they relied on a single assessment for each text rather than multiple independent annotators. This opens risk of inconsistent quality, annotator bias, and insufficient inter-annotator reliability measurement. For a benchmark whose core variable is language, translation quality control needs to be stronger (random audits, multiple annotators, kappa statistics, or quantitative MT metrics).

3. The authors use LLMs (GPT variants) both to create/standardize task descriptions and to translate. That raises two concerns: (a) the dataset creation process could inadvertently incorporate phrasing that favors certain LLMs (e.g., GPT family) and (b) the benchmark might not be neutral if authors used models that overlap with models under test. The authors partly mitigate by keeping 6 private tasks and manual review, but the risk should be acknowledged and quantified more thoroughly.

4. It’s unclear whether prompts, temperature, decoding choices, and instruction templates were controlled or tuned per language. LLM performance is often prompt-sensitive; different languages might require language-specific prompt engineering. If the same prompt template (translated mechanically) was used, that may disadvantage non-English languages. The paper should document prompt templates and any language-specific prompt adaptation (or explicitly justify not doing so).

5. Six private tasks help with benchmark-to-pretraining leakage, but six may not be enough to make general claims about un-seen tasks; the authors themselves recommend expanding private tasks.

6. Translators/annotators performed unpaid validation. This raises concerns and potential quality/reliability issues. For a benchmark of broad impact, the annotation process should be reproducible and ethical (compensated, multiple annotators, quality checks).

7. Important experimental details (complete list of tasks, per-task results, prompt templates) are relegated to the Appendix

8. The dataset construction depends solely on Kaggle competitions. While Kaggle offers well-structured ML problems, this narrow source might constrain the diversity of domains, task formulations, and data modalities included. Many real-world ML workflows differ substantially from Kaggle settings. It would strengthen the paper to justify this design choice and discuss whether the benchmark’s conclusions generalize beyond Kaggle-style supervised learning tasks.

**Questions:**

1. How were the 30 Kaggle competitions (24 public + 6 private) selected, and what measures were taken to ensure diversity across domains, data modalities, and task types? Given the exclusive reliance on Kaggle, how do the authors assess the benchmark’s generalizability beyond Kaggle-style supervised ML problems?

2. Since translations were generated with GPT-4o and validated by a single native-speaker annotator per text, how do the authors ensure translation consistency and reliability across languages? Were any inter-annotator checks, audits, or quantitative MT quality metrics (e.g., BLEU, COMET) performed or planned?

3. Because GPT-family models were used both to standardize task descriptions and to translate them, how do the authors mitigate potential bias that could favor similar models in evaluation? Have alternative translation or standardization pipelines been tested to confirm benchmark neutrality?

4. Were prompt templates, decoding settings, and temperature parameters controlled or tuned per language? If identical English templates were directly translated, how do the authors ensure linguistic equivalence and fairness given prompt sensitivity across languages?

5. Do the six private tasks provide sufficient coverage for assessing pretraining leakage, and are there plans to expand them? Additionally, could the authors clarify the ethical considerations of unpaid annotation work and confirm whether full task lists, prompt templates, and evaluation artifacts will be publicly released for reproducibility?

**Details Of Ethics Concerns:**

Unpaid annotators

---

> ### Author Response · Authors · 2025-12-04
>
> We would like to thank the reviewer for their valuable and constructive feedback. We have prepared detailed responses addressing their questions and clarifying several points. (The reviewer’s original comments are presented in quotes.)
>
> > How were the 30 Kaggle competitions (24 public + 6 private) selected, and what measures were taken to ensure diversity across domains, data modalities, and task types?
>
> Based on the received feedback, we expanded our benchmark to 35 competitions: we retained 22 public competitions and incorporated the remainder from private ones. The 22 public competitions were selected from Code4ML~2.0, which covers competitions up to 2024. We then added 3 new competitions from 2025, bringing the total number of public competitions to 25—the release year of this paper. All competitions were selected according to the following criteria:
>
> * The competition is closed, so the data and leaderboard are fixed.
> * The dataset is downloadable from the Kaggle page.
> * The evaluation metric is documented on the competition page.
> * The evaluation is reproducible, with all necessary metadata provided.
> * The Kaggle submission format is a tabular prediction.
>
> We aimed to maintain diversity across domains, modalities, and task types. This was more challenging for private competitions due to their smaller pool, but we continue to work toward improving diversity across the above dimensions.
>
> > How do the authors ensure translation consistency and reliability across languages? Were any inter-annotator checks, audits, or quantitative MT quality metrics (e.g., BLEU, COMET) performed or planned?
>
> Constructing a multilingual benchmark requires substantial resources and effort. We acknowledge that using at least three native speakers for validation in each language would have been more reliable. However, we initially operated under the assumption that GPT-4 already provides high-quality translations (see section \textit{Translating Data}), and that validation by a trusted and proficient native speaker would correct any remaining issues.
>
> Following the reviewer’s suggestion, we included a detailed analysis of BLEU scores. The mean BLEU scores across three runs exceed 0.4 (typically considered a satisfactory level) for all languages and text types except Kazakh data-description texts. Moreover, the intervals for descriptions and data cards are relatively narrow, suggesting stable and accurate translations.
>
> For domain-level texts, the confidence intervals are notably wider. This is expected, as BLEU is highly sensitive to short texts, and GPT-4o may produce less consistent translations for brief inputs lacking contextual cues.
>
> Overall, the BLEU results support our claim that single-annotator validation produced translations of sufficient quality for the present study. Detailed confidence-interval plots are included in the revised paper.
>
> > Translators/annotators performed unpaid validation. This raises concerns about quality and reliability
>
> As noted above, because we relied on single-annotator validation, selecting high-quality native speakers was essential; most annotators were fellow researchers closely associated with the authors. Although the validation work was unpaid, we had strong confidence in their professionalism and expertise, as they were trusted colleagues with a demonstrated track record of careful and reliable work. Consequently, we do not expect the lack of financial compensation to have adversely affected annotation quality.
>
> > Do the six private tasks provide sufficient coverage for assessing pretraining leakage, and are there plans to expand them?
>
> To better assess pretraining leakage, we expanded the benchmark to include four additional private tasks (for a total of ten). We plan to continue increasing both the number and diversity of private tasks in future iterations to further strengthen leakage-robust evaluation.

---

### Note · Authors · 2026-01-06

I have read and agree with the venue's withdrawal policy on behalf of myself and my co-authors.